# The Inflammatory Contribution of B-Lymphocytes and Neutrophils in Progression to Osteoporosis

**DOI:** 10.3390/cells12131744

**Published:** 2023-06-29

**Authors:** Drew Frase, Chi Lee, Chidambaram Nachiappan, Richa Gupta, Adil Akkouch

**Affiliations:** 1Western Michigan University Homer Stryker M.D. School of Medicine, Kalamazoo, MI 49008, USA; drew.frase@wmed.edu (D.F.);; 2Department of Orthopaedic Surgery and Medical Engineering Program, Western Michigan University Homer Stryker M.D. School of Medicine, Kalamazoo, MI 49008, USA

**Keywords:** osteoporosis, B-lymphocytes, neutrophils, osteoimmunology, bone remodeling, Bregs, TGF-β1, IL-10, inflammation, aging

## Abstract

Osteoporosis is a bone disease characterized by structural deterioration and low bone mass, leading to fractures and significant health complications. In this review, we summarize the mechanisms by which B-lymphocytes and neutrophils contribute to the development of osteoporosis and potential therapeutics targeting these immune mediators to reduce the proinflammatory milieu. B-lymphocytes—typically appreciated for their canonical role in adaptive, humoral immunity—have emerged as critical regulators of bone remodeling. B-lymphocytes communicate with osteoclasts and osteoblasts through various cytokines, including IL-7, RANK, and OPG. In inflammatory conditions, B-lymphocytes promote osteoclast activation and differentiation. However, B-lymphocytes also possess immunomodulatory properties, with regulatory B-lymphocytes (Bregs) secreting TGF-β1 to restrain pathogenic osteoclastogenesis. Neutrophils, the body’s most prevalent leukocyte, also contribute to the proinflammatory environment that leads to osteoporotic bone remodeling. In aged individuals, neutrophils display reduced chemotaxis, phagocytosis, and apoptosis. Understanding the delicate interplay between B-lymphocytes and neutrophils in the context of impaired bone metabolism is crucial for targeted therapies for osteoporosis.

## 1. Introduction

Osteoporosis, a disorder characterized by reduced bone mass and the structural deterioration of osteogenic tissue, affects a significant proportion of people in the United States over 50 years old [1]. Considering a total of almost 10 million cases and an estimated 30 million at risk, osteoporosis is an ever-growing health concern [1]. In a meta-analysis conducted by Salari et al., using 86 studies across five countries, the global osteoporosis prevalence was 18.3%, with high rates observed in European and African countries [2]. In the United States, the age-adjusted prevalence of adults 50 years and older was almost 13% from 2017 to 2018, with an increased prevalence in men compared to women (19.6% and 4.4%, respectively) [3].

Osteoporosis most commonly manifests in the form of femoral neck or vertebral fractures, which can significantly reduce quality of life [4]. Fracture consequences include disability, extensive medical costs, and even increased mortality—underscoring the importance of understanding the mechanisms of osteoporosis and potential therapeutics [5]. In addition to direct physical consequences, the fiscal burden of osteoporosis and its concomitant fractures—both directly through therapeutic expenses and indirectly through missing school or work—impose substantial challenges to United States healthcare systems [6,7]. Economic forecasts suggest that by 2025, yearly fractures from osteoporosis will eclipse 3 million cases at a predicted cost of 25.3 billion dollars [8,9].

Recent research has begun to highlight the immune system’s critical role in the development and progression of osteoporosis, focusing mainly on soluble mediators, growth factors, and chemokines [10]. Emerging work has implicated both adaptive and innate immune system components, paying particular attention to their contribution to proinflammatory milieu [11,12,13]. In reference to these findings, Srivasta et al. proposed a novel term, “immunoporosis,” to stress the importance of immune cells in osteoporosis pathogenesis [14,15].

The pathophysiology of T-lymphocytes in osteoporosis is studied, but the exact role of B-lymphocytes and neutrophils as inflammatory mediators needs to be clarified. In this paper, we offer a comprehensive review of the immunomodulatory mechanisms connecting B-lymphocytes and neutrophils to the development of osteoporosis and therapies aimed at reducing B-lymphocyte- and neutrophil-induced inflammation.

## 2. B-Lymphocytes

### 2.1. B-Lymphocytes and Bone Homeostasis in Osteoporosis

B-lymphocytes canonically represent the most significant component of the humoral adaptive immune system. After developing from hematopoietic stem cell (HSC) precursors in the bone marrow, mature B-lymphocytes’ primary role is to secrete antibodies that neutralize pathogens and potentiate the effector functions of other immune cells. B-lymphocytes play a direct role in antigen-dependent T-cell activation in the lymph node. Beyond their traditional roles in adaptive immunity, however, B-lymphocytes are increasingly being considered for their potential roles in diseases of bone remodeling. Given the anatomical proximity of the bone architecture to immune cell genesis in the bone marrow, osteoblasts, osteoclasts, and immune cells have long been thought to communicate via shared signaling mechanisms. Recent terms such as “osteoimmunology” and “immunoporosis” have explored the relationship between bone homeostasis and immune cells in osteoporosis-related diseases. However, little attention has been paid to the pathogenic disruptions to bone homeostasis driven specifically by B-lymphocytes [14,15].

In the bone marrow, B-lymphocyte development is highly dependent upon secreted factors derived from stromal cells and osteoblasts. During B-lymphocyte maturation, successful V(D)J recombination generates an immunoglobulin heavy chain (Ig-H-chain) as part of the pre-B-cell receptor (pre-BCR) [16]. Bruton’s tyrosine kinase (BTK) signaling appears to attenuate signals through the pre-BCR to limit proliferation [16]. Contrastingly, signaling through mature B-lymphocyte receptors (BCRs) leads to the activation of the phosphorylation of tyrosine residues on BTK [17]. Surprisingly, BTK signaling also plays a critical role in osteoclast differentiation [18,19]. Mice deficient in BTK signaling exhibited an osteopetrosis-like phenotype due to impaired bone resorption from deficient RANKL:BTK signaling in osteoclasts [18]. Additionally, when X-linked immunodeficient mice—with dysfunctional BTK signaling—were treated with receptor activator of nuclear factor kB (NF-Kb), osteoclast precursors failed to fuse into active multinucleated osteoclasts [19].

Over the last few decades, various studies have linked key cytokines and molecules to B-lymphocyte development and bone remodeling, including IL-7, receptor activator of nuclear factor kB (RANK), and osteoprotegerin [OPG]. Initial IL-7/IL-7R knockout studies observed that IL-7 transgenic mice had increased B-lymphocyte precursor levels, leading to a purported link between defects in B-lymphocyte development and bone mass [20]. Similar mouse studies knocking out RANK—the receptor for RANKL—led to fewer B-lymphocytes in lymph nodes [21]. RANK, RANKL, and OPG play a pivotal role in the differentiation and activation of osteoclasts. Shortly after IL-7 and RANK experiments began associating B-lymphocytes with bone remodeling, B-lymphocytes were shown to secrete OPG (Figure 1) [22]. OPG is a tumor necrosis factor (TNF) receptor family member; it is a molecular decoy receptor that binds RANKL, inhibiting RANKL:RANK-mediated osteoclastogenesis and preventing excessive bone resorption. Surprisingly, B-lymphocytes have been shown to produce roughly half of the total OPG produced in the bone marrow [23]. In mice with B-lymphocytes knocked out, their bone marrow was deficient in OPG, and osteoporotic bone was more prevalent than in controls [24]. Significantly, B-lymphocyte transplantation rescued the osteoporotic phenotype and improved bone marrow OPG levels [24]. In addition to osteoclastogenic suppression, B-lymphocytes also inhibit osteoblast differentiation via the secretion of C-C motif chemokine ligand 3 (CCL3) and TNF, which target extracellular signal-regulated kinase (ERK) and nuclear factor kappa-light-chain-enhancer of activated B cells (NF-kB) to impair osteoblast differentiation [25].

The role of B-lymphocytes in bone remodeling is context-dependent. B-lymphocytes secrete many factors critical in maintaining the bone architecture and share many cytokines with osteoclasts and osteoblasts. While B-lymphocytes play a critical role in suppressing osteoclasts via the secretion of OPG receptor decoys, inflammatory environments can redirect B-lymphocyte effects on bone remodeling toward bone resorption (Figure 1) [26]. In particular, IL-6, TNF-α, and IL-1β, are critical in driving the inflammatory pathophysiology of bone remodeling diseases [27]. In inflammatory conditions, B-lymphocytes have been shown to secrete RANKL, which stimulates the activation of osteoclasts [28,29,30]. Comparatively, RANKL knockout mice experienced greater protection against bone loss following ovariectomy than controls, while knockout in T-cells did not protect against bone loss after ovariectomy [30].

In addition to RANKL-mediated osteoclast activation, B-lymphocytes in inflammatory environments also secrete granulocyte colony-stimulating factor (G-CSF), which leads to osteoclast progenitor proliferation [31]. The production of both G-CSF and RANKL drives the proliferation and differentiation of osteoclasts, ultimately leading to bone resorption and loss of bone mass. G-CSF also appears to play a role in regulating neutrophil infiltration, which can lead to enhanced inflammation [31]. B-lymphocytes have been known to secrete IL-18 for years; however, the effect of IL-18 on bone remodeling was first reported to be anti-osteoclastogenic [32]. B-lymphocyte secretion of IL-18 was initially shown to upregulate OPG expression on osteoblastic cells, inhibiting osteoclastogenesis through the OPG/RANKL axis [32]. In a more recent study, however, B-lymphocyte secretion of IL-18 increased the surface expression of RANKL on T-lymphocytes, ultimately contributing to osteoclastogenesis [33]. In the latter study of osteoporotic women, IL-18 was increased compared to controls.

The contrasting roles of B-lymphocytes in osteoclastogenesis support the crucial role of B-lymphocytes in the balance of metabolic bone remodeling. Thus, disruptions of B-lymphocyte-mediated signaling via inflammatory environments are thought to drive catabolic bone resorption by promoting osteoclast differentiation and activation. Osteoporosis, in particular, appears to be driven in part by disrupted B-lymphocyte and bone homeostasis. In one postmenopausal osteoporosis study, women with osteoporosis had significantly fewer CD19+ B-lymphocytes than healthy controls [26]. Healthy controls also secreted less macrophage colony-stimulating factor (M-CSF) and had increased bone mineral density (BMD) in their lumbar spine [26]. Despite the association, it remains unclear whether reduced BMD impairs B-lymphocyte development or if impaired B-lymphocyte development reduces BMD.

### 2.2. Regulatory B-Lymphocytes (Bregs) in Osteoporosis

In addition to traditional antibody-secreting B-lymphocytes, B-regulatory lymphocytes (Bregs) are being increasingly recognized for their immunomodulatory role in bone homeostasis. Various Breg cytokines, including TGF-β1, IL-10, and IL-35 have been shown to modulate osteogenic differentiation. IL-35 is a ligand for the IL-35 receptor, which drives Breg differentiation via STAT1/STAT3 signaling pathways [34]. IL-35:IL-35-R binding also appears to suppress osteoclastogenesis by OPG secretion and subsequent RANKL downregulation [35].

Breg regulatory functions are largely attributed to the anti-inflammatory cytokine IL-10 [36,37]. Of note, CD19+ CD1dhi CD5+ [B10] Bregs were recently shown to secrete IL-10 and protect against osteoclastogenic pathogenesis [38]. In a recent animal model, IL-10 induced osteoblast differentiation by downregulating miR-7015-5p [39]. Besides driving osteoblast differentiation, IL-10 also suppresses osteoclast development. IL-10 directly impairs Ca^2+^ mobilization and NFATc1 signaling in osteogenic precursors, preventing the development of osteoclasts [40]. Furthermore, the adoptive transfer of a subset of Breg (B10) cells appeared to delay the onset of osteoporosis in an ovariectomized mouse model [41].

TGF-β1 is also a well-recognized anti-inflammatory cytokine and is also secreted by Bregs [42,43]. TGF-β1 and bone morphogenic proteins (BMP) constitute a critical developmental axis that signals traditionally via SMADS or nontraditionally via p38 MAPK pathways [42,43]. While the roles of TGF-β1 as a developmental morphogen are receiving increasing attention, fewer studies document the anti-inflammatory benefits of TGF-β1 in osteoporosis. The SMAD and MAPK pathways converge on signaling cascades that upregulate pro-osteoblastic factors such as Runt-related transcription factor 2 (RUNX2) [42]. As with IL-10, TGF-β1 also appears to decrease NFAT signaling and decreases RANK expression on osteoblasts [42]. TGF-β1 also decreases the expression of crucial osteoclastic genes cathepsin K and acid phosphatase 5, tartrate-resistant [42]. Thus, in osteoporotic conditions where TGF-β1 is reduced, RANKL expression would remain on osteoblasts, and RUNX2-directed osteoblastic differentiation would be reduced, both of which shift bone remodeling toward resorption.

Bregs represent an exciting new frontier in the field of osteoimmunology and immunoporosis. Currently, the anti-inflammatory mechanisms of Bregs, mediated via IL-10 and TGF-β1, offer promise in tempering proinflammatory environments that increase osteoclastogenesis and promote the pathogenic bone loss characteristic of osteoporosis. Current studies showing the positive protective effects of Bregs in bone remodeling are limited to animal models but have nonetheless been promising. Particularly in osteoporosis, it is imperative to shift the metabolic balance away from resorption.

## 3. Neutrophils

Neutrophils are polymorphonuclear granulocytes. They are the most abundant leukocytes in the body, representing 40–60% of leukocytes in the blood. Several cytokines, such as G-CSF and GM-CSF, control neutrophil proliferation and differentiation. Mature neutrophils are essential in multiple parts of the innate immune system [44]. Mature neutrophils are rapidly recruited to the site of inflammation in response to cytokines and chemokines, leading to adhesion to endothelial cells. At sites of inflammation, neutrophils can then synthesize chemokines C-X-C and C-C to eliminate pathogens via phagocytosis, chemical degranulation, and the extrusion of neutrophil extracellular traps (NETs). Neutrophils were also recently discovered to play a role in adaptive immunity regulation [45]. An excellent example of neutrophils playing this dual role occurs in osteoarthritis. Neutrophils are the first cells to enter the synovium as part of the innate immune system. Their presence in the synovium continues as neutrophils play a critical role in tissue degeneration via chemokine and cytokine secretion and other enzymes’ activation, changing osteoblast and osteoclast activity [46].

### 3.1. Neutrophils Are Affected by Aging

As humans age, there is a decrease in the immune response, termed immunosenescence. One part of the immune response affected by senescence is neutrophils. Studies have confirmed that chemokinesis is unchanged in neutrophils, but chemotaxis is reduced due to reduced chemotactic signaling [47,48]. Neutrophil phagocytosis is aided by opsonization by antibodies, complement receptor CD35, or Fc receptors CD16 and CD32. It has been found that there is an age-related reduction in the number of pathogens phagocytosed. Decreased phagocytosis is associated with an age-related decline in CD16 surface expression [49].

Free radical production is affected in aged individuals due to a signaling cascade. Neutrophils are typically stimulated by formyl-methionyl-leucyl-phenylalanine (fMLP) to produce a burst of reactive oxygen species (ROS). This function with changes in age is still being studied. Some studies have found that in the older age groups, there is a decrease in ROS within the first 18 h but then an unexplained increase in ROS after 48 h compared to younger age groups. Neutrophils have NADPH oxidase (NOX2), which produces superoxide. The NOX2-derived superoxide enhances osteoclast differentiation to promote osteoporosis by upregulating a downstream mediator called nuclear factor of activated T cells c1 (NFATc1) [50]. Interestingly, the pro-osteoclast factor RANKL also increased ROS levels through NADPH-mediated mechanisms [51]. Other studies have noticed a decrease in ROS in response to Gram-positive infections, and the results hint at multiple pathways affecting neutrophil activation [48,49,52,53]. Thus, aged individuals with increased systemic inflammation and ROS will favor a metabolic shift toward osteoclastogenesis, and their neutrophils will be less effective at generating ROS bursts to combat infection.

There is an apparent reduction in neutrophil function and phenotype, but, surprisingly, the relative abundance of neutrophils increases in aged individuals. The other significant change in neutrophils with aging is the number of immune cells. A decrease in immune function theoretically corresponds to a decrease in circulating neutrophils. However, relative neutrophil abundance increases with age. The number of circulating neutrophils increases while the number of neutrophils undergoing differentiation and proliferation remains the same. Increased neutrophil abundance occurs because of a decrease in the function of the apoptosis pathway (Figure 2). GM-CSF, IL-2, and lipopolysaccharide (LPS) challenge can rescue neutrophils from apoptosis. However, GM-CSF is implicated in rescuing neutrophils from apoptosis in aged individuals, potentially due to GM-CSF playing a vital role in the ratio of Bax/Mcl-1. In younger individuals, Mcl-1 usually overcomes Bax levels to induce apoptosis, but in aged individuals, the Jak2-STAT5 signal transduction pathway can be impaired, leading to increased Bax expression [54].

### 3.2. Neutrophil Function in Regard to Osteoporosis

Neutrophil senescence plays a role in multiple pathologies, osteoporosis being one of them. There have been several proposed mechanisms to explain the hypothesized relation between neutrophil senescence and impaired bone metabolism. One exciting idea includes the predictive nature of the neutrophil-to-lymphocyte ratio (NLR). As discussed earlier, there is an increase in circulating neutrophils in aged individuals due to an alteration in the apoptosis rescue pathway. When neutrophils fail to undergo apoptosis, they accumulate, leading to an increased NLR associated with osteoporosis [55].

Neutrophils can lead to augmented bone loss in osteoporosis patients. Those who have postmenopausal osteoporosis have decreased estrogen. Estrogen influences neutrophil activation, chemotaxis, and ROS generation in addition to the general decline in function found in aging neutrophils [45]. Without estrogen, fewer neutrophils mobilize to the sites of inflammation, resulting in Th17 cells secreting more IL-17, inducing osteoclastic bone resorption. Neutrophils express RANKL and IL-8 in certain conditions to activate osteoclastogenesis. RANKL expression by neutrophils is induced by Toll-like receptor 4 (TLR4) activation [46]. Neutrophils are also known to adhere to osteoblasts to induce osteoblast retraction and stimulate bone resorption [45].

Neutrophils can also affect bone homeostasis by modulating the fate decisions of mesenchymal stem cells (MSCs). While the increased number of activated neutrophils influences MSC differentiation into osteoblasts, it was also discovered that neutrophils inhibit extracellular matrix secretion by MSCs. G-CSF-induced neutrophil proliferation can result in the apoptosis of MSCs and osteoblasts via ROS [45].

Another potential contributor to the pathogenesis of osteoporosis includes the dysfunction of neutrophil homeostatic function. Neutrophils typically have an adhesion molecule such as LAD-1 for extravasation to an inflammatory site, where macrophages release chemokine IL-23 for neutrophil recruitment [44]. With a defective LAD-1 or related adhesion molecule, neutrophils cannot enter the site of inflammation. Without neutrophils being allowed to enter, the macrophages will continue to release IL-23, triggering IL-17 production from T cells. IL-17 leads to inflammatory bone loss [56].

Neutrophil deficiency is another factor that can lead to osteoporosis. Gfi1 is a key molecule involved in hematopoiesis development. GFI1 mutations lead to severe congenital neutropenia. A study using Gfi1 knockout mice found that exposure to a pathogen led to low bone mass, whereas those without pathogen exposure were unaffected. Depending on the pathogen load and amount of systemic inflammation, osteoporosis can occur through changes in homeostasis, favoring osteoclastogenesis [57].

In summary, neutrophil senescence and dysfunction can affect osteoclasts, osteoblasts, and MSCs (directly or indirectly), contributing to osteoporosis and other issues, such as osteoarthritis [46]. Further studies should aim to clarify the impaired molecular processes in neutrophils that contribute to inflammatory environments in aged individuals [45].

## 4. Therapeutics

The first key component in the prevention and treatment of osteoporosis is calcium supplementation. Calcium is an essential nutrient, critical for bone health, as it is a significant component of bone tissue and is necessary for bone mineralization [58]. For many years, calcium supplementation has been recommended as a critical component of osteoporosis prevention and treatment, along with vitamin D and lifestyle modifications.

### 4.1. Calcium and Vitamin D

Several studies have assessed the efficacy of calcium supplementation in the prevention and treatment of osteoporosis. A meta-analysis conducted by Liu et al. (2020) found that calcium supplementation was associated with a significant increase in BMD compared to a placebo in postmenopausal women [58]. Another randomized controlled trial (RCT) led by Yu et al. (2012) showed improved BMD and a decreased fracture risk in postmenopausal women supplemented with calcium compared to controls [59].

However, not all studies have found that isolated calcium supplementation positively affects bone health. Additionally, it is crucial to consider supplementation with vitamin D, a nutrient critical for calcium absorption and utilization [60]. Randomized controlled trials analyzed by Weaver et al. (2015) showed that vitamin D supplementation and the use of calcium led to a significant reduction in risk for toral or hip fractures (15% and 30% reduction, respectively) [61]. However, an excess of vitamin D has also been discussed as potentially harmful to bone. In another randomized clinical trial, Burt et al. (2019) found that treatment with 4000 IU to 10,000 IU vitamin D units per day for three years resulted in a statistically significantly lower radial bone mineral density in the radius bone for healthy adults when compared to treatment with only 400 IU units of vitamin D per day [62]. The study also revealed a statistically significantly lower tibial bone mineral density in the tibial bone amongst the patients who underwent a treatment regimen of vitamin D at 10,000 IU compared with 400 IU per day.

The benefits of vitamin D extend beyond direct calcium absorption and include beneficial anti-inflammatory effects. It is well known that the vitamin D receptor (VDR) is expressed on B-lymphocytes and T-lymphocytes [63]. Additionally, one study found increased anti-inflammatory IL-10 cytokine secretion from dendritic cells stimulated with vitamin D [64]. Thus, the dual benefits of increased calcium absorption and immunomodulation suggest a protective role for vitamin D in maintaining BMD. Monitoring bone mineral density is crucial in preventing osteoporosis, since the uncontrolled activity of osteoclasts will more likely affect weakened bones and increase the susceptibility to fractures. Calcium and vitamin D are considered natural and adequate preventative therapeutic options for osteoporosis. However, bisphosphonates have emerged as the treatment for osteoporosis in most developing countries.

### 4.2. Bisphosphonates

Bisphosphonates have a similar structure to pyrophosphate, the natural regulator of calcium homeostasis in the body [65]. This structural analog allows bisphosphonates to inhibit osteoclast-mediated bone resorption, commonly seen in osteoporosis conditions, and they do so by disrupting protein trafficking and normal cell function pathways in osteoclasts [65]. Bisphosphonates are the treatment of choice for osteoporosis because they reduce bone loss, increase bone mineral density, and significantly decrease the risk of fractures. Some examples of bisphosphonates include alendronate (Fosamax), risedronate (Actonel), teriparatide, ibandronate (Boniva), and zoledronic acid (Reclast) [66]. A study by Naylor et al. (2016) examined an exciting parameter: gauging effective bone production via surrogate markers of bone remodeling, using bone turnover markers to signal increased bone production in alendronate, ibandronate, and risedronate treatments. Both C-terminal telopeptide of type 1 collagen (CTX) and procollagen type I N-propeptide (PINP) had an increased magnitude in 70% of the women in the study, regardless of which bisphosphonate treatment was administered [66]. Bisphosphonate use in combination with other immune therapies offers an exciting avenue for treatment.

### 4.3. Combination Therapy with Bisphosphonates

#### 4.3.1. Bisphosphonates and Monoclonal Antibodies

In addition to the bisphosphonate treatment discussed above, combining bisphosphonate with monoclonal antibodies, such as denosumab and romosozumab, has proven to be an effective therapy. Denosumab interrupts the binding of RANKL to RANK, ultimately dampening osteoclast-mediated bone resorption [67]. Romosozumab behaves in a similar protein-targeting fashion, inhibiting sclerostin to simultaneously promote bone formation while limiting bone resorption [68]. A randomized controlled trial by Tsai et al. (2013) placed postmenopausal women with osteoporosis in three separate groups: treatments with teriparatide alone, treatments with denosumab alone, and treatments with both. The study showed a significant increase in BMD in the lumbar spine, femoral neck, and total hip with combination therapy, compared to a single treatment with denosumab or teriparatide [69]. Another randomized controlled trial conducted by Saag et al. (2017) tested a combination treatment of romosozumab/alendronat in postmenopausal women with osteoporosis. The study showed that romosozumab/alendronate combination therapy led to a nearly 50% reduction in the risk of vertebral fractures and a 19% decrease in nonvertebral fractures [68]. Indeed, the combination therapy of romosozumab administration prior to alendronate resulted in a significantly reduced fracture risk amongst postmenopausal osteoporotic patients compared to bisphosphonate (alendronate) treatment alone.

Rituximab is a monoclonal antibody targeting the B-lymphocyte CD20 receptor. Rituximab has primarily been used to treat autoimmune diseases such as rheumatoid arthritis (RA), which are driven by B-lymphocyte pathogenesis [70]. The aforementioned role of B-lymphocytes in the progression of osteoporosis suggests a potential role for rituximab therapy in osteoporosis. However, in one study of rituximab therapy in women with RA, one year of treatment did not have an effect on BMD [71]. Conversely, another retrospective study found significantly increased BMD in vertebral sections L1–L4 after 18 months of rituximab therapy [72]. The role of rituximab in preventing BMD loss should be studied for extended treatment regimens to determine the true long-term efficacy. Furthermore, combination therapy with rituximab and bisphosphonates should also be explored as potential therapeutics for metabolic bone diseases such as osteoporosis.

#### 4.3.2. Bisphosphonates and Parathyroid Hormone

Another combination therapy with bisphosphonate is the addition of parathyroid hormone (PTH) analogs. Parathyroid hormone is a hormone released in the body that helps to control calcium levels in the blood [69]. Parathyroid hormone accomplishes this by stimulating calcium release from the bones into the bloodstream in a feedback loop [67]. Abaloparatide, an anabolic agent and a parathyroid hormone analog medication, has been used to treat osteoporosis [66]. A clinical trial conducted by Bone et al. (2018) on postmenopausal women with osteoporosis examined a 2-year regimen (24 months) of alendronate and 18 months of abaloparatide before alendronate administration. After the treatment period, the study measured vertebral fractures, nonvertebral fractures, and relevant clinical fractures, showing a marked reduction with alendronate/abaloparatide treatment compared to a placebo [65]. The study observed a decrease in the risk of osteoporotic fractures and an increase in BMD with 18 months of abaloparatide treatment prior to the administration of alendronate [61]. PTH treatments should consider the context-dependent effect of PTH on bone metabolism. Short, or daily, dosing of PTH directs osteoblasts to form new cortical bone, while constitutive PTH dosing—exhibited in hyperparathyroidism—drives catabolic osteoclast resorption [73].

### 4.4. Hormone Therapy

Another treatment for osteoporosis is hormone therapy, especially for postmenopausal women. Hormone therapy involves using estrogen or estrogen–progesterone combinations to replace the declining hormones after menopause. As mentioned above, estrogen is found to influence neutrophil activation and chemotaxis. Estrogen also plays a key role in recruiting neutrophils to sites of inflammation, resulting in Th17 cells secreting more IL-17, inducing osteoclastic bone resorption. In a meta-analysis by Prior et al. of randomized controlled trials that sampled more than 1000 menopausal women receiving estrogen alone and estrogen–progesterone treatments, the study found a significant increase in BMD over one year [74]. By supplementing the body with estrogen or estrogen–progesterone combinations, hormone therapy can help to reestablish bone metabolism equilibrium and reduce the fracture risk in postmenopausal women.

### 4.5. Electrical Stimulation

Another treatment for osteoporosis is an invasive approach in the form of electrical stimulation. Functional electrical stimulation combines the benefits of both electrical stimulation and mechanical loading. The technique involves the application of a constant direct current to the repair site by implanting a cathode and anode [75]. Animal studies have experimentally confirmed that electrical stimulation with direct currents has a therapeutic effect on osteoporosis. In a study by Iimura et al., the researchers demonstrated that electrical stimulation of the superior laryngeal nerve fibers in rats resulted in calcitonin secretion [76]. Iimura et al. also found a statistically significant increase in BMD in the tibial and femoral metaphysis in rats under chronic electrical stimulation versus the control group [76]. In another study conducted by Yuen-Chi Lau et al., the researchers applied electrical stimulation to the dorsal root ganglion in rats. They found similar secretion of calcitonin gene-related peptides that suppress osteoporosis [77]. One intriguing study used varying electrical stimulation frequencies to stimulate a pro- or anti-inflammatory macrophage phenotype [78]. Although the benefits of electrical stimulation for BMD are currently under investigation, more work should continue to leverage electrical stimulation therapy to modify immune cell function.

### 4.6. Strontium

Strontium is a mineral that has been studied as a potential treatment for osteoporosis. Strontium is thought to have a dual effect on bone health by inhibiting bone resorption and stimulating bone formation [79]. Strontium affects osteoblasts by increasing the expression of osteogenic genes such as Runx2, ALP, BSP, and BGP [80]. Strontium also affects osteoclasts by inhibiting osteoclast precursors’ differentiation and promoting osteoclast apoptosis [80]. A study by Zhao et al. looked at strontium implants in osteoporotic rabbits and observed their interactions. The researchers observed hydrophilicity and the activation of osteointegration under osteoporotic states. These included the regeneration of the bone tissue surrounding the implant, high biocompatibility, and an improvement in osteogenic abilities [81]. In vitro studies have been performed with animal models to examine strontium-enriched hydroxyapatite bioceramics, while studies with human subjects have yet to be well defined. One promising in vitro study coated titanium oxide with various ratios of calcium and strontium and was able to polarize macrophages away from the inflammatory M1 phenotype and toward the anti-inflammatory, TGF-β-producing, M2 phenotype [82]. As with electrical stimulation, the direct effects of strontium therapy on immune cell function should continue to be explored.

### 4.7. Wnt and Cathepsin K

In a review article by Khosla et al. (2017), the researchers suggest that the understanding and manipulation of molecular pathways such as Wnt signaling could be crucial to bone mass homeostasis [83]. Future research in antibodies and Wnt inhibitors, such as DKK-1 in rodent models, has been explored as a potential option [83]. Khosla et al. also discussed cathepsin K, a catabolic protein secreted by mature osteoclasts to degrade bone matrix products such as type 1 collagen [83]. In addition, Khosla et al. mentioned that clinical trials are underway for oral cathepsin K inhibitors, and this treatment option may leave osteoclasts and osteoblasts intact by targeting the product that is secreted [83]. In an RA mouse model, knockout of cathepsin K not only led to the preservation of bone but also a signification decrease in inflammation [84]. Further research should consider the anti-inflammatory effects of cathepsin K inhibition in addition to the direct benefit of osteoclast impairment.

### 4.8. Current Treatment Limitations and Future Directions

While several effective treatments are available for osteoporosis, they are not without their potential side effects. These side effects can range from mild discomfort to severe complications that significantly impact a patient’s quality of life. Therefore, it is essential to understand the potential side effects of osteoporosis treatments to make informed decisions about the best course of action in managing this disease. For daily calcium supplementation in adults, the Institute of Medicine recommends 1000–1200 mg [85]. Excessive calcium supplement intake can lead to side effects such as gas, constipation, and bloating [85]. Furthermore, excessive calcium intake can lead to hypercalcemia and increase the risk of kidney stones and cardiovascular disease [85]. A prospective study from Li et al. (2012) attempted to associate calcium intake and supplementation with acute cardiovascular issues such as stroke or even myocardial infarction. The researchers observed a statistically significantly increased risk of myocardial infarction in users of calcium supplements. However, no association was found for those obtaining calcium from the diet, such as dairy products [86]. Overall, the study suggested that increasing dietary calcium does not lead to significant cardiovascular complications, but calcium supplements should be taken cautiously [86]. Vitamin D has been deemed “generally safe” by medical institutions such as the Mayo Clinic. However, excessive vitamin D in the form of supplements can harm vulnerable populations such as children, pregnant women, and older adults [87]. Symptoms that accompany excessive vitamin D supplementation (more than 4000 IU units daily) include kidney damage, cardiovascular complications, constipation, weakness, poor appetite, and disorientation [87].

Bisphosphonates are now one of the more prominent treatment options for osteoporosis, but the treatment can also be associated with harmful effects. A study conducted by researchers from the Mayo Clinic discussed adverse effects such as gastroesophageal irritation and osteonecrosis of the jaw that can be observed from bisphosphonate treatment [88]. Gastroesophageal irritation can result in severe conditions such as esophageal ulcers or inflammation. In a study conducted with romosozumab and denosumab monoclonal antibody treatment for patients with rheumatoid arthritis accompanied by osteoporosis by Mochizuki et al. (2021), the researchers observed side effects such as rash, joint pain, nausea, headache, hypertension, and itching in patients administered romosozumab and denosumab [89]. Abaloparatide, the parathyroid hormone analog used in combination with bisphosphonate, can also exhibit side effects, such as constipation, loss of appetite, pain in the bone and joints, and weakness [90].

Future research in osteoporosis treatment may focus on several key areas.

There is a need for a better understanding of the underlying molecular mechanisms of bone loss and the factors contributing to osteoporosis development. This knowledge could inform the development of new drugs that target these pathways.Better methods to diagnose osteoporosis early, when treatment is most effective, can be beneficial. Advances in imaging techniques, such as high-resolution CT scans and MRI, may provide more accurate measures of bone density and help to identify individuals at risk of developing osteoporosis.The development of new and innovative treatment strategies that can improve bone health and reduce the risk of fractures may include gene therapy, stem cell transplantation, and novel drug therapies that target specific pathways involved in bone formation and resorption.

## 5. Conclusions

The consideration of osteoporosis within the context of our immune system has widened the landscape of treatment avenues for osteoporosis. Traditional treatment has focused on calcium and bisphosphonates—the side effects of which have been previously mentioned. New strategies should focus on engineered approaches that inhibit the proinflammatory environments that drive osteoclastogenesis. IL-10 is a desirable target, as it is shown to act through the downregulation of a specific microRNA, miR-7015-5p [35]. microRNA technology can be rapidly designed and personalized, making it an attractive therapy avenue. TGF-β1 should also be explored further as a potential anti-inflammatory mediator in bone metabolism due to its ability to decrease osteoclastic genes [38]. TGF-β1 and IL-10 are secreted by Breg cells and can inhibit the differentiation of osteoclasts and preserve bone mass. Additionally, researchers should further explore the idea of the adoptive transfer of Bregs, shown previously to delay osteoporosis onset in a mouse model [29]. The adoptive transfer of Bregs, if safe, would provide the host with both TGF-β1 and IL-10, which could exert a powerful dampening effect—via NFAT-related signaling mechanisms—on inflammatory environments, where osteoclasts predominate. Targeting GM-CSF to restore the Bax/Mcl-1 ratio in aged neutrophils is an attractive avenue for therapy. Despite their functional decline, aged neutrophils have reduced apoptosis, increasing NLR. Targeting Jak2-STAT5 in the Bax/Mcl-1 signaling axis to restore the NLR in aged patients should be explored further. As with B-lymphocytes, the primary goal in osteoporotic treatment is to restore bone metabolism to equilibrium. New technologies focusing on electrical stimulation and strontium therapy should also receive increasing attention as therapies to combine with additional approaches. Wnt and cathepsin K inhibitors are also attractive targets, and their evaluation as candidate therapies should not be overlooked. By focusing on novel therapies that leverage the reciprocal communication between our immune system and developing bone tissue, we can continue to advance treatments for people diagnosed with osteoporosis.

## Figures and Tables

**Figure 1 cells-12-01744-f001:**
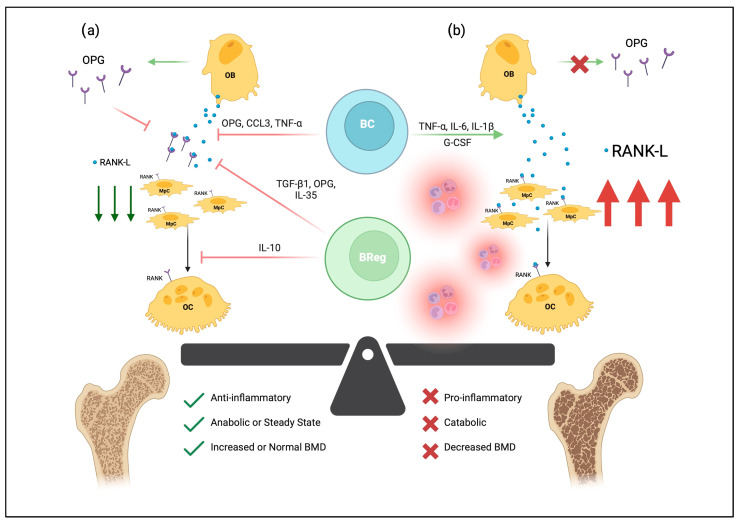
The effects of B-lymphocytes on bone homeostasis. B-lymphocytes (BC) and osteogenic precursors develop together in the bone marrow niche. In (**a**), osteoblasts (OB) can secrete RANKL, which binds RANK on the surfaces of monocyte-like progenitor cells (MpC) and stimulates differentiation and fusion into multinucleated osteoclasts (OC). RANKL also binds RANK on osteoclasts to promote survival and proliferation. In healthy bone, pro-osteoclastogenic RANKL is regulated by the osteoprotegerin (OPG) decoy receptor, as well as various soluble mediators secreted by OBs, BCs, and B-regulatory lymphocytes (Bregs). Thus, normal B-lymphocyte and osteogenic signaling generates an anti-inflammatory environment that maintains bone mineral density (BMD). In osteoporotic tissue, seen in (**b**), proinflammatory mediators drive osteoclastogenesis through increased expression of RANK-L, shifting the balance of bone metabolism toward catabolism and ultimately bone loss. Created with BioRender.com, accessed on 3 June 2023.

**Figure 2 cells-12-01744-f002:**
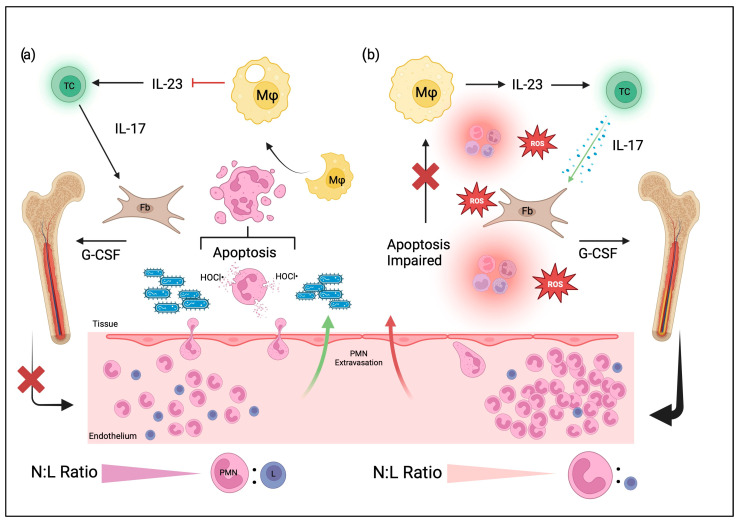
Impaired tissue apoptosis in aged neutrophils triggers excessive neutrophil mobilization. (**a**) Normal neutrophils (polymorphonuclear, PMN) extravasate at the endothelial lining into the tissue, release a burst of bleach (HOCl•) to fight bacterial infection, and eventually undergo apoptosis. Macrophages (Mp) phagocytose apoptotic neutrophils and deactivate IL-23 secretion, which decreases IL-17 and granulocyte colony-stimulating factor (G-CSF) signaling and pauses neutrophil mobilization from the bone marrow. Thus, the neutrophil-to-lymphocyte ratio (NLR) is maintained with normal neutrophil function. In aged individuals with greater systemic inflammation and ROS, shown in (**b**), neutrophils have impaired chemotaxis, extravasation, and apoptosis. Macrophages are then free to secrete IL-23, which increases IL-17 expression from TH17 T-lymphocytes (TC) and G-CSF from fibroblasts (fb). Increased G-CSF leads to increased neutrophil release from the bone marrow and a high NLR ratio. Created with BioRender.com, accessed on 3 June 2023.

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
