# Peer review of "The Inflammatory Contribution of B-Lymphocytes and Neutrophils in Progression to Osteoporosis"

_cells, 2023, doi:10.3390/cells12131744_

Round 1
Reviewer 1 Report
Figure 1. incorrect figure and description. osteoblasts [OB] and osteogenic precursors [OpC] do not differentiate into osteoclasts (OC). Monocyte-like progenitor cells are osteoclast progenitor cells. OBs are completely distinct from OCs. There is no RANK on the surface of OBs in Fig1. a) and b). Please change the figure and the description appropriately.
Author Response
Point 1: Figure 1. incorrect figure and description. osteoblasts [OB] and osteogenic precursors [OpC] do not differentiate into osteoclasts (OC). Monocyte-like progenitor cells are osteoclast progenitor cells. OBs are completely distinct from OCs. There is no RANK on the surface of OBs in Fig1. a) and b).
Response 1: We agree that the figure needs to be modified. Both figure 1a) and 1b) no longer have RANK on the OB surface. RANK has been moved to the surface of MpCs and OCs, along with the addition of Ob-secreted RANKL and OPG. We believe the figure more accurately shows how secreted RANKL can drive oseteoclastogenesis from Monocyte-like progenitor cells. The caption was updated to match the updated figure.
An earlier reference to ‘Figure 1’ was added to paragraph 2 of section 2.1 (line 82).
Reviewer 2 Report
The review 'The inflammatory contribution of B-lymphocytes and neutrophils in progression to osteoporosis' by Frase et al shows that B cells and neutrophils contribute to the proinflammatory environment that leads to osteoporotic bone remodeling. The review is well written but has the following minor issues.
1. Please describe the role of BTK signaling in B cells with regards to osteoporosis as it is one the most important signaling kinase for B cell function.
2. Please also describe the role oxidants in B cells/neutrophils with respect to osteoporosis.
3. Role of ROS from neutrophils is not incorporated in Fig. 2.
4. If B cells have inflammatory functions, will anti-B cells therapy like Rituximab be effective in Osteoporosis. Please discuss it in section of therapeutics.
Only minor editing required
Author Response
Point 1: Please describe the role of BTK signaling in B cells with regards to osteoporosis as it is one the most important signaling kinase for B cell function.
Response 1: We have provided an updated description of the role of BTK in B-lymphocytes and acknowledge the surprising necessity of BTK in osteoclasts. While BTK signaling has been known to play roles in immunodeficiencies (XLA), leukemias, and lymphomas, there is not currently any literature specifically focusing on the targeting of BTK for osteoporosis. We choose to place more emphasis on the secreted soluble factors that influence RANKL levels, rather than the intrinsic signaling mechanism of B-cells. We believe the addition of BTK signaling in B-lymphocyte and osteoclast is sufficient.
Point 2: Please also describe the role oxidants in B cells/neutrophils with respect to osteoporosis.
Response 2: The role of ROS was expanded upon in relation to osteoclastogenesis. It’s role in B-cells is less clear and not well studied currently. Therefore, we thought it best to keep the discussion of ROS in the section on neutrophils.
Neutrophils have NADPH oxidase (NOX2) which produce superoxide. The NOX2 derived superoxide enhances osteoclast differentiation to promote osteoporosis by upregulating a downstream mediator called nuclear factor of activated T cells c1 (NFATc1) [E]. While it has not been studied, it is likely that the decreased ROS burst in aged neutrophils is more detrimental to clearing infection than beneficial for bone metabolism. Interestingly, RANKL also increased ROS levels through NADPH-mediated mechanisms [F].
Point 3: Role of ROS from neutrophils is not incorporated in Fig. 2.
Response 3: Figure 2a was updated to include the MClO- released by neutrophils in tissue. The aged side, figure 2b was updated to reflect greater systemic inflammation and ROS. Additionally, the caption was updated to reflect the figure changes.
Point 4: If B cells have inflammatory functions, will anti-B cells therapy like Rituximab be effective in Osteoporosis. Please discuss it in section of therapeutics.
Response 4: We agree that a section talking about rituximab should be included. In section 4.3, subsection ‘Bisphosphonates and monoclonal antibodies’, a section detailing rituximab effects on BMD was included. We believe the inclusion of rituximab strengthens the treatment section of the paper and helps tie back to the previously mentioned role of B-lymphocytes in section 2.
Reviewer 3 Report
The work is of a high standard, highlighting the relevance of the problem. However, there are some minor remarks that should have been corrected.
1) Line 30, the word osteoporosis should be written with a small letter
2) In each of the sub-paragraphs in the Therapeutics chapter, the therapeutic approaches and their impact on the immune system should be linked, conceptually expand these sub-paragraphs
Author Response
Point 1: Line 30, the word osteoporosis should be written with a small letter
Response 1: The change was incorporated in line 30 of the updated manuscript.
Point 2: In each of the sub-paragraphs in the Therapeutics chapter, the therapeutic approaches and their impact on the immune system should be linked, conceptually expand these sub-paragraphs
Response 1: We accept the recommendation to expand the therapeutic subsections to incorporate their impact on the immune system. The sections on vitamin D, PTH, electrical stimulation, strontium therapy, and cathepsin K were all updated and tied back to their impact on the immune system.